

# Respiratory syncytial virus, human metapneumovirus, and influenza virus infection in Bangkok, 2016-2017

Ilada Thongpan, Nungruthai Suntronwong, Preeyaporn Vichaiwattana, Nasamon Wanlapakorn, Sompong Vongpunsawad and Yong Poovorawan

Center of Excellence in Clinical Virology, Department of Pediatrics, Faculty of Medicine, Chulalongkorn University, Bangkok, Thailand

## ABSTRACT

Children and adults residing in densely populated urban centers around the world are at risk of seasonal influenza-like illness caused by respiratory viruses such as influenza virus, human metapneumovirus (hMPV), and respiratory syncytial virus (RSV). In a large metropolitan of Thailand's capital city Bangkok, most respiratory infections are rarely confirmed by molecular diagnostics. We therefore examined the frequency of RSV, hMPV, and influenza virus in 8,842 patients who presented influenza-like illness and sought medical care at a large hospital in Bangkok between 2016 and 2017. Using a multiplex real-time reverse-transcription polymerase chain reaction (RT-PCR), 30.5% (2,699/8,842) of nasopharyngeal (NP) swab samples tested positive for one or more of these viruses. Influenza virus comprised 17.3% (1,528/8,842), of which the majority were influenza A/H3N2. Such infection was most prevalent among adults and the elderly. RSV was identified in 11.4% (1,011/8,842) and were mostly ON1 and BA9 genotypes. Of the hMPV-positive samples (3.6%, 318/8,842), genotypes A2, B1, and B2 were detected. A small number of individuals experienced co-infections (1.8%, 155/8,842), most commonly between RSV and influenza A/H3N2. RSV and hMPV co-infections were also found, but mainly in young children. Viral respiratory tract infection peaked locally in the rainy season (June to September). These findings support the utility of rapid nucleic acid testing of RSV, hMPV, and influenza virus in patients with ILI.

Corresponding author
Yong Poovorawan,
Yong.P@chula.ac.th

## INTRODUCTION

Respiratory tract infection is a major contributor to morbidity and mortality among children and adults worldwide (*Boloursaz et al., 2013*; *Garg et al., 2015*). Most recognized is the seasonal influenza virus infection, which is responsible for about 290,000 to 650,000 deaths each year (*WHO, 2018*). Epidemiological studies have shown that infants, young children, and the elderly are especially at risk of infection by both subtypes of RSV (designated A and B) (*Henrickson et al., 2004*). Even hMPV is now recognized as a frequent cause of acute respiratory tract infections in children predominantly ≤5 years of age, elderly adults, and immunocompromised patients (*Johnstone et al., 2008*; *Williams et al., 2004*).

Each of the two genetically and antigenically distinct groups of hMPV (A and B) can be further divided into genetic subgroups 1 and 2 (*Boivin et al., 2002*).

Multiple groups of different respiratory viruses frequently co-circulate with a variable pattern of predominance in Thailand. Data on the prevalence of infection caused by these viruses are often incomplete and limited due to their similar clinical presentation and seasonality overlap (*Thanasugarn et al., 2003*; *Horthongkham et al., 2014*). The systematic use of molecular diagnostics such as the real-time reverse transcription-polymerase chain reaction (RT-PCR) assay has been important in improving accurate diagnosis of viral respiratory infections and has proven extremely useful for disease surveillance (*Mahony, 2008*).

Here, we aimed to assess the disease burden caused by RSV, hMPV, and influenza virus in a large patient population of all ages who presented influenza-like illness (ILI) and sought medical care at a hospital in Bangkok within the past two years.

## MATERIALS AND METHODS

### Study design and specimens

We retrospectively tested 8,842 stored respiratory samples obtained from both in-patient and out-patient individuals of all ages with ILI who sought medical care at Bangpakok 9 International Hospital in Bangkok and collected consecutively between January 2016 and December 2017. ILI was defined as fever (>38 °C) and accompanying respiratory symptoms such as cough, sore throat, or pharyngitis. This study analyzed de-identified convenient samples and extended an earlier investigation of an ongoing influenza virus prevalence in Thailand (*Suntronwong et al., 2017*). Available patient information included gender and age, but not extensive clinical information nor disease severity. The Institutional Review Board of the Faculty of Medicine of Chulalongkorn University approved this study (IRB number 609/59).

### Real-time RT-PCR

RNA was extracted from 200 μL of specimens using the Viral Nucleic Acid Extraction Kit (RBC Bioscience, Taiwan, R.O.C.) according to the manufacturer's instructions. RSV and hMPV detections were performed using an in-house TaqMan-based multiplex one-step real-time RT-PCR. The primers and probes targeted the M gene of RSV and the F gene of hMPV (Table 1). The RSV probe was labeled with 6-carboxy-fluorescein (FAM) at the 5′ end and Black Hole Quencher-1 (BHQ-1) at the 3′ end. The hMPV probe was labeled with 6-carboxy-fluorescein (HEX) at the 5′ end and Black Hole Quencher-1 (BHQ-1) at the 3′ end. The reaction mixture contained 2 μL RNA, 10 μmol of each of the primers and probes, and SensiFAST Probe No-ROX One-Step reagent (Bioline, London, UK). Cycling parameters included 1 cycle for 20 min at 42 °C, initial denaturation for 3 min at 95 °C, 50 cycles for 10 s at 95 °C and 20 s at 60 °C. This assay has a limit of detection of 100 genome copies per reaction for both viruses. No cross-detections were observed between the two viruses and other respiratory viruses including influenza A and B viruses, adenovirus, enterovirus, rhinovirus, and coronavirus. Real-time RT-PCR assay of influenza A and B viruses was previously described (*Suwannakarn et al., 2008*). Parallel detection of
**Table 1  Primers and probes used to detect RSV, hMPV, and influenza virus.**

|  | Virus | Primer/Probe | Sequence 5′–3′ | Target | Position | |
|---|---|---|---|---|---|---|
| Assay 1 | RSV A and B | RSV_F3251 | GGCAAATATGGAAACATACGTGAA | M | 3251-3274 | (+) |
|  |  | RSV_R3334 | TCTTTTTCTAGGACATTGTAYTGAACAG | M | 3334-3361 | (-) |
|  |  | RSV_P3303 | FAM-CTGTGTATGTGGAGCCTTCGTGAAGCT-BHQ1 | M | 3303-3329 | (+) |
|  | hMPV A and B | hMPV_F3604 | CAARTGYGACATTGCTGAYCTRAA | F | 3604-3628 | (+) |
|  |  | hMPV_R3683 | ACTGCCGCACAACATTTAGRAA | F | 3683-3662 | (-) |
|  |  | HMPV_P3630 | JOE-TGGCYGTYAGCTTCAGTCARTTC-BHQ1 | F | 3630-3643 | (+) |
| Assay 2[a] | Influenza A | FluA-M-F151 | CATGGARTGGCTAAAGACAAGACC | M | 151-175 | (+) |
|  |  | FluA-M-R276 | AGGGCATTTTGGACAAAKCGTCTA | M | 276-252 | (-) |
|  |  | FluA-M-P218 | FAM-ACGCTCACCGTGCCCAGT-BHQ1 | M | 218-235 | (+) |
|  | Influenza B | FluB-MF439 | CTCTGTGCTTTRTGCGARAAAC | M | 439-460 | (+) |
|  |  | FluB-MR | CCTTCYCCATTCTTTTGACTTGC | M | 671-649 | (-) |
|  |  | FluB-P135 | Cy5-TCAGCAATGAACACAGCAA-BHQ3 | M | 541-559 | (+) |
|  | Influenza A/H1N1 | H1_F | ACTACTGGACTCTGCTKGAA | H1 | 750-769 | (+) |
|  |  | H1_R | AAGCCTCTACTCAGTGCGAA | H1 | 846-827 | (-) |
|  |  | H1_P | FAM-TTGAGGCAAATGGAAATCTAATAGC-TAMRA | H1 | 789-813 | (+) |
|  | Influenza A/H3N2 | H3_F | TGCTACTGAGCTGGTTCAGAGT | H3 | 139-160 | (+) |
|  |  | H3_R | AGGGTAACAGTTGCTGTRGGC | H3 | 322-302 | (-) |
|  |  | H3_P | HEX-AGATGCTCTATTGGGAGACC-BHQ1 | H3 | 226-245 | (+) |
|  | GAPDH | GAPDH-F85 | GTGAAGGTCGGAGTCAACGG | GAPDH | 85-104 | (+) |
|  |  | GAPDH-R191 | TCAATGAAGGGGTCATTGATGG | GAPDH | 191-169 | (-) |
|  |  | GAPDH-P121 | HEX-CGCCTGGTCACCAGGGCTGC-BHQ1 | GAPDH | 121-140 | (+) |

**Notes.**

[a]Previously described in *Suwannakarn et al. (2008)*.

(+) and (-) denote sense and anti-sense strand, respectively.

glyceraldehyde 3-phosphate dehydrogenase (GAPDH) gene served as an internal control. Fluorescence signals cycle threshold (Ct) was based on optimization and values ≤38 were considered positive.

## Conventional RT-PCR

Samples tested positive for RSV and/or hMPV were genotyped. Complementary DNA was synthesized using the ImProm-II Reverse Transcription System (Promega, Madison, WI, USA) according to the manufacturer's instructions. RNA and random hexamers were incubated at 70 °C for 5 min, followed by extension for 2 h at 42 °C and inactivation at 70 °C for 15 min. Amplification of the partial RSV glycoprotein (G) gene inclusive of the second hypervariable region (HVR2) and the F gene was performed using semi-nested RT-PCR as previously described (*Auksornkitti et al., 2014*). Cycling parameters were initial denaturation at 94 °C for 3 min, 40 cycles of denaturation at 94 °C for 20 s, annealing at 55 °C for 20 s, elongation at 72 °C for 90 s, and a final extension at 72 °C for 10 min. Identical amplification parameters were carried out in the second-round PCR for 30 cycles. Partial F-gene of hMPV was subjected to nested-PCR as previously described (*Chung et al., 2008*). The PCR conditions were initial denaturation at 95 °C for 3 min, 35 cycles of 95 °C for 1 min, 55 °C for 1 min, 72 °C for 1 min, and a final extension at 72 °C for 3 min. The

PCR products for RSV-A (840 bp), RSV-B (720 bp), and hMPV (750 bp) were visualized using 2% agarose gel electrophoresis and purified using the GeneAll Expin gel extraction kit (GeneAll Biotechnology, Seoul, South Korea) according to the manufacturer's instructions. Purified PCR products were subjected to Sanger sequencing.

### Sequence and phylogenetic analyses of RSV and hMPV genotypes

Nucleotide sequences of RSV and hMPV strains were aligned using ClustalW implemented in BioEdit (version 7.0.9) by comparison to the sequences previously assigned to specific genotypes (Table S1 and Table S2). Phylogenetic trees were constructed using the maximum likelihood method implemented in the MEGA6 (*Tamura et al., 2013*). The reliability of the tree based on the Tamura–Nei model was estimated using 1,000 bootstrap pseudo-replications. Sequences were considered the same genotype if they clustered together with bootstrap values of 70–100% (*Venter et al., 2001*).

Nucleotide sequences were submitted to the GenBank database under the accession numbers MH447703–MH447725 (RSV-A), MH447726–MH447818 (RSV-B), and MH447819–MH447950 (hMPV).

### Statistical analysis

The association between virus prevalence and the patient age at infection was assessed using univariate analyses (SPSS software version 22.0). $P$-values were calculated using the Chi-squared test or Fisher's exact test, where cell counts below 5 were used. A $p$-value of <0.05 was considered statistically significant.

## RESULTS

### The overall prevalence of RSV, hMPV, and influenza virus

We retrospectively tested 8,842 consecutive respiratory samples (48.5% males, age range 0–106 years). Of these, 30.5% (2,699/8,842) were positive for one or more viruses. Influenza virus was most commonly identified (17.3%, 1,528/8,842), followed by RSV (11.4%, 1,011/8,842) and hMPV (3.6%, 318/8,842) (Table 2). Influenza virus and RSV were more prevalent in 2016 than in 2017. To facilitate analysis, samples were categorized into seven groups in order to examine the distribution of viral infection relative to age (Table 3 and Table S3). Regardless of gender, the burden of RSV was greatest among children 5 years of age and younger (21.2% and 15.4% among those ≤2 and 3-5 years of age, respectively). Frequency of RSV infection appeared to decrease with increasing age and was <9% in those older than 5 years of age ($p < 0.0001$). In contrast, influenza virus infection was more frequently found among older individuals. Meanwhile, hMPV infection was distributed among all ages (1.7–5.7%).

### Seasonal and genotype distribution of RSV, hMPV, and influenza virus

The prevalence of viral etiology of influenza-like infection differed slightly among the viruses examined. Among 1,011 RSV-positive samples, subgroup identification was possible for 488 specimens. Of these, 36.3% (177/488) were RSV-A and 66.4% (324/488) were RSV-B. RSV infection appeared most frequently in the rainy months (between July and November) with

**Table 2 Overall prevalence of samples tested positive for RSV, hMPV or influenza virus.**

| Year | No. of samples | Virus-positive samples (%) | RSV-positive (%) | hMPV-positive (%) | Influenza virus-positive (%) |
|------|---------------|---------------------------|------------------|-------------------|------------------------------|
| 2016 | 4,178 | 1,428 (34.2) | 590 (14.1) | 114 (2.7) | 814 (19.5) |
| 2017 | 4,664 | 1,271 (27.3) | 421 (9.0) | 204 (4.4) | 714 (15.3) |
| Total | 8,842 | 2,699 (30.5) | 1,011 (11.4) | 318 (3.6) | 1,528 (17.3) |

**Table 3 Characteristics of samples and detection frequency of RSV, hMPV, and influenza virus.**

| Characteristics | | Samples (%) ($N = 8,842$) | RSV (%) ($N = 1,011$) | hMPV(%) ($N = 318$) | Influenza A+B (%) ($N = 1,528$) |
|-----------------|---|---------------------------|------------------------|----------------------|----------------------------------|
| Age, year (mean ± SD age) | ≤2 (1.2 ± 0.6) | 1,916 (21.7) | 406 (21.2) | 105 (5.5) | 134 (7.0) |
| | 3–5 (3.8 ± 0.8) | 1,541 (17.4) | 238 (15.4) | 88 (5.7) | 160 (10.4) |
| | 6–12 (8.4 ± 1.9) | 1,253 (14.2) | 100 (8.0) | 25 (2.0) | 298 (23.8) |
| | 13–18 (15.2 ± 3.2) | 371 (4.2) | 19 (5.1) | 10 (2.7) | 101 (27.2) |
| | 19–30 (25.4 ± 3.2) | 1,148 (13.0) | 66 (5.7) | 20 (1.7) | 211 (18.4) |
| | 31–60 (41.4 ± 8.1) | 2,164 (24.5) | 144 (6.7) | 53 (2.4) | 516 (23.9) |
| | >60 (72.0 ± 9.1) | 449 (5.1) | 38 (8.5) | 17 (3.8) | 108 (24.1) |
| *p*-value | | | **0.0262** | 0.5695 | **<0.0001** |
| Gender | Male | 4,288 (48.5) | 514 (50.8) | 160 (50.3) | 742 (48.6) |

**Notes.**
Statistically significant differences among groups are bolded.

the highest annual prevalence of 37% (206/555) and 17.3% (136/784) in August 2016 and September 2017, respectively (Fig. 1A). Although RSV-A and RSV-B were equally detected in 2016, RSV-B was more frequently identified in 2017. From 318 hMPV-positive samples, subgroup identification was possible for 132 specimens. Of these, 80.3% (106/132) were hMPV-B, which was the predominant subgroup in both years (Fig. 1B). From 1,528 samples tested positive for influenza virus, there were more influenza A virus (76.2%, 1,164/1,528) than influenza B virus (23.8%, 364/1,528). In 2016, high prevalence of influenza virus occurred twice, 20.5% (59/288) in March and 34.5% (234/678) in September (Fig. 1C). The following year, peak influenza virus activity occurred in August (25.3%, 185/732). Overall, A/H3N2 accounted for 70% (815/1,164) of all influenza A virus.

## Genotyping and phylogenetic analysis of RSV and hMPV

Partial G gene sequences that were randomly selected to identify the RSV genotypes showed all of the RSV-A strains (23/23) were genotype ON1 and all of the RSV-B strains (93/93) were genotype BA9 (Figs. 2A and 2B, respectively). Inter-subgroup diversity between A_ON1 and B_BA9 was relatively high (p-distance value of 2.17–2.44). In contrast, genetic variations among intra-genotype strains were relatively small (*p*-distance value of 0–0.073 and 0–0.071 within the ON1 and BA9 genotypes, respectively).

Partial F gene sequences were obtained from 132 of the 318 hMPV-positive specimens. Phylogenetic analysis of 132 hMPV strains identified in this study showed two main genetic lineages, A and B. Strains clustered into subgroup A2, B1, and B2, but not subgroup A1 (Fig. 3). The majority of the strains belonged to subgroup B1 (74%, 98/132), while only 6% (8/132) belonged to subgroup B2. The remaining 20% of the strains (26/132) belonged to

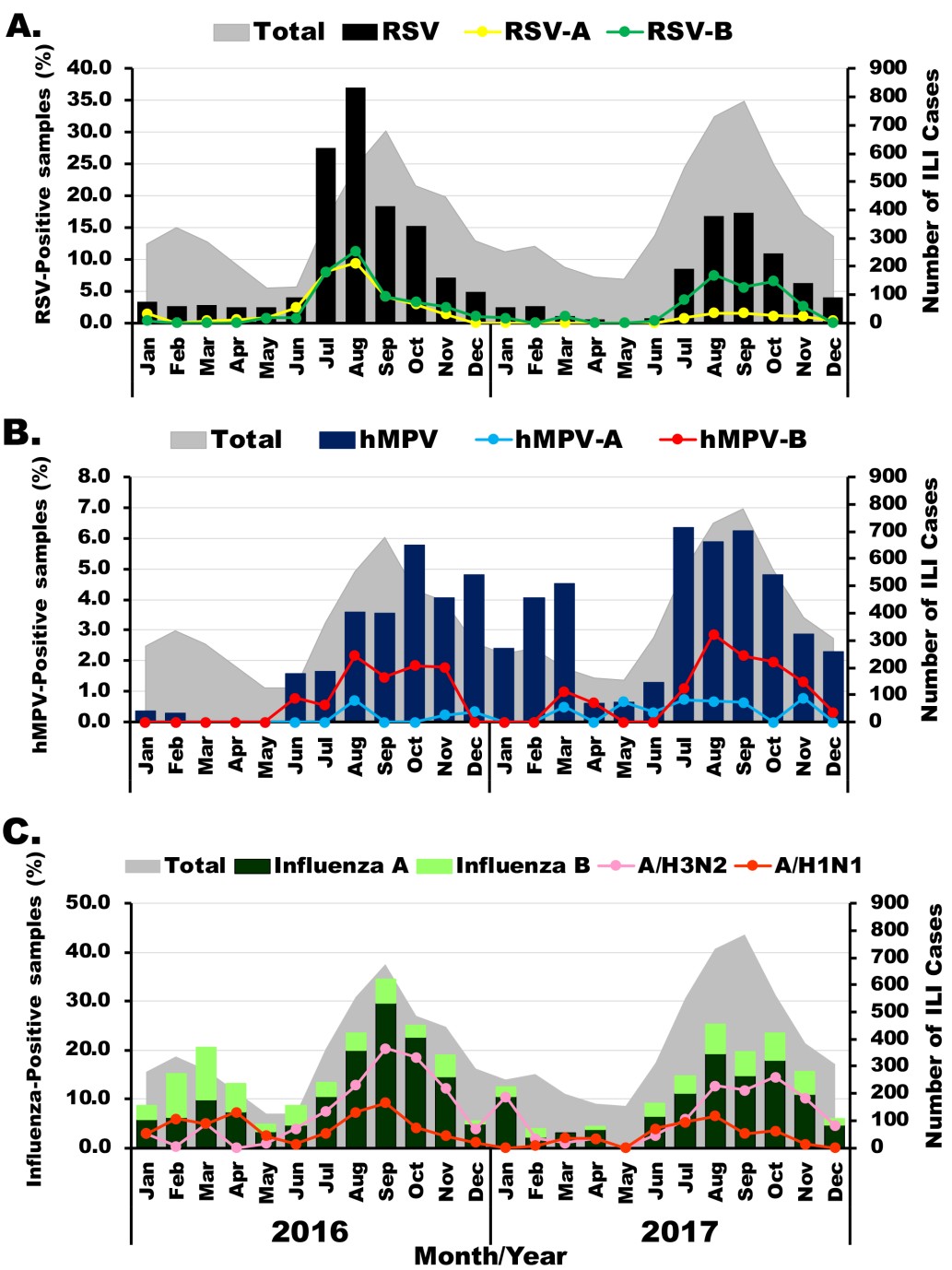

**Figure 1 Seasonal distribution of infection for each virus.** The monthly number of samples from patients with influenza-like illness (ILI) is shown in gray (right scale). (A) Bar graphs show RSV-positive rate with RSV-A in yellow and RSV-B in green (left scale). (B) Bar graphs show hMPV-positive rate with hMPV-A in blue and hMPV-B in red (left scale). (C) Bar graphs show frequency of influenza A (dark green) and influenza B (light green) virus infection with A/H1N1 in orange and A/H3N2 in pink (left scale).

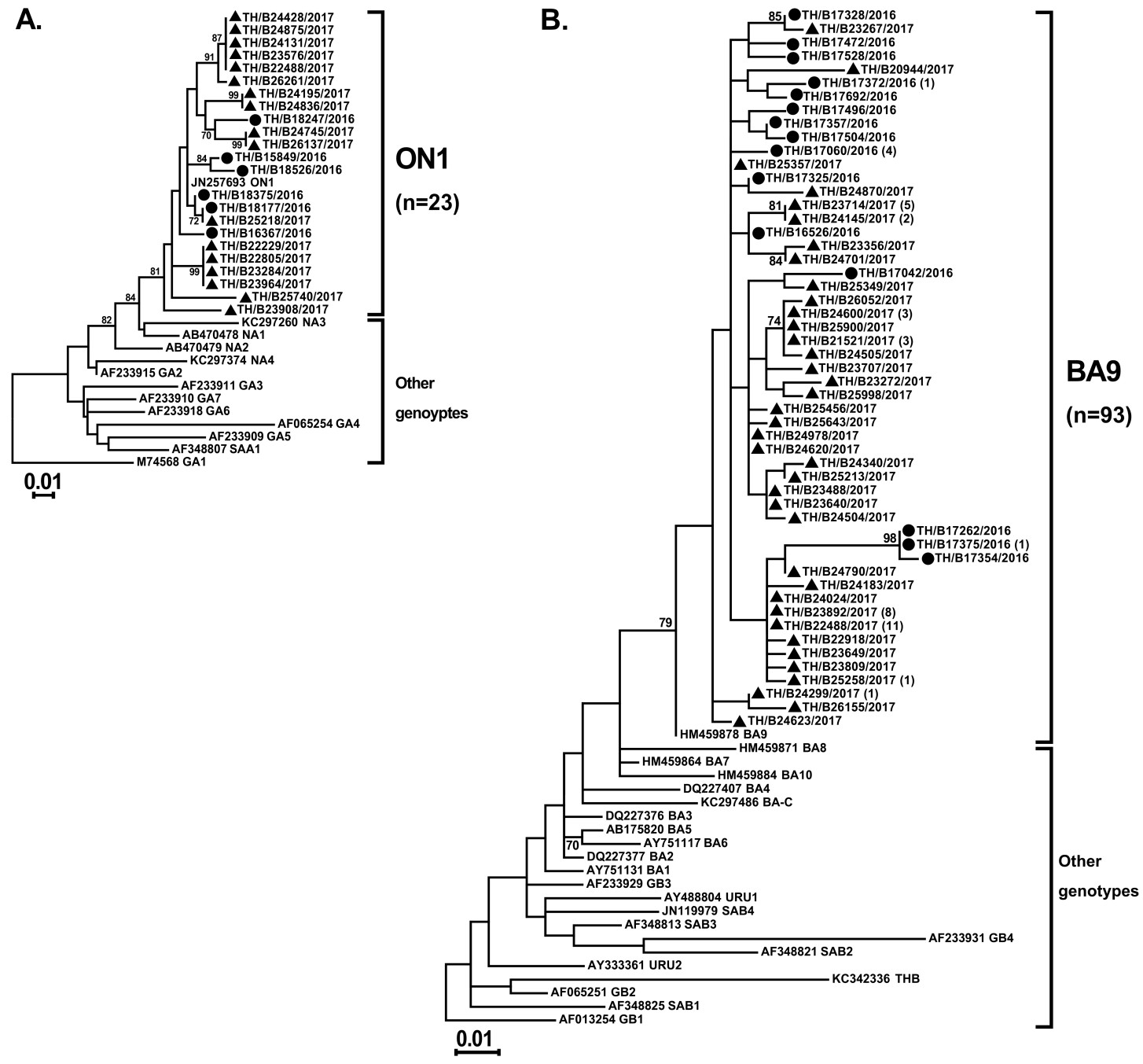

**Figure 2 Phylogenetic analysis of RSV subgroup A and B based on the nucleotide sequence encompassing the HVR2 region within the G gene.** Trees were constructed using the maximum likelihood method based on the Tamura–Nei model and implemented in MEGA6. Bootstrap values of 1,000 pseudo-replications > 70% are indicated at the branch nodes. Reference sequences for each genotype (GA1–GA7, SAA1, NA1–NA4, and ON1 for RSV-A and GB1–GB4, SAB1–SAB4, URU1, URU2, THB, and BA1–BA10 for RSV-B) were obtained from GenBank. The scale bar represents the number of nucleotide substitutions per site between close relatives. Circles denote samples from Thailand 2016, while squares indicate strains from Thailand 2017 The number of strains are shown in parentheses.

subgroup A2. The intra-genotype strains were genetically closely related (*p*-distance values of 0.001–0.019), while the inter-subgroup comparisons were more diverse (*p*-distance values of 0.087–0.116).

### Coinfections among RSV, hMPV, and influenza virus

The frequency of single versus multiple infections and the number of co-occurrences of viruses for each possible virus combination were examined (Table 4). The most common combination observed was RSV(non-typed) and influenza A H3N2 subtype ($n = 68$). As a percentage, the virus most often found in coinfections was RSV, which was found in 17.4% (176/1,011) of the samples, followed by hMPV (10.4%, 33/318), and influenza virus (8.5%, 130/1,528).

## DISCUSSION

This study was conducted over a two-year study period between 2016–2017 among 8,842 patients who presented with influenza-like infections. Two multiplex real-time reverse transcriptase polymerase chain reaction (RT-PCR) assays were used to rapidly detect three of the most common viral respiratory pathogens. It was not surprising that influenza was the most prevalent virus (17.3%), followed by RSV (11.4%) and hMPV (3.6%). Similar to our findings, previous study examining hospitalized patients with lower respiratory tract infections in Thailand found that influenza viruses were the most common respiratory viruses diagnosed among ILI cases (*Chittaganpitch et al., 2018*). RSV prevalence was highest among children aged <5 years with rates of infection between 15.4 and 21.2%. On the other hand, RSV had a lower burden of symptomatic respiratory illness among older children and adults, and the opposite trend was observed for influenza virus infection. The proportion of patients with influenza virus infections increased with age, and the rate of infection was greatest in children 13–18 years of age (27.2%). These findings are supported by previously reported studies on the epidemiology of respiratory virus infection (*Zhang et al., 2014*; *Richter et al., 2016*).

In the present study, the seasonal distribution of influenza virus infections resembled those of RSV and hMPV infections, which was similar to data from previous studies (*Richter et al., 2016*; *Parsania et al., 2016*; *Chittaganpitch et al., 2018*). Although Thailand is located geographically in the northern hemisphere, the seasonality of respiratory infection is similar to that of several nearby tropical settings such as Indonesia, Malaysia, the Philippines and the Southern hemisphere countries of Australia and New Zealand (*Weber, Mulholland & Greenwood, 1998*; *Paynter et al., 2015*). In these regions, respiratory infections generally peak in the rainy season and declines during the hot and dry months. Moreover, a study from Bangladesh found an increased risk of respiratory infection following rainy days, suggesting a link between rainfall and population crowding or proximity (*Murray et al., 2012*). In Thailand, the period when students are in school overlaps with the rainy season, so it is possible that host behavior is associated with an increased risk of respiratory infection.

In our study, both RSV subgroups A and B circulated during the same RSV season, but the relative proportions varied as subgroup B occurred more frequently than subgroup A in the 2017 season. Several previous studies including from our group have reported

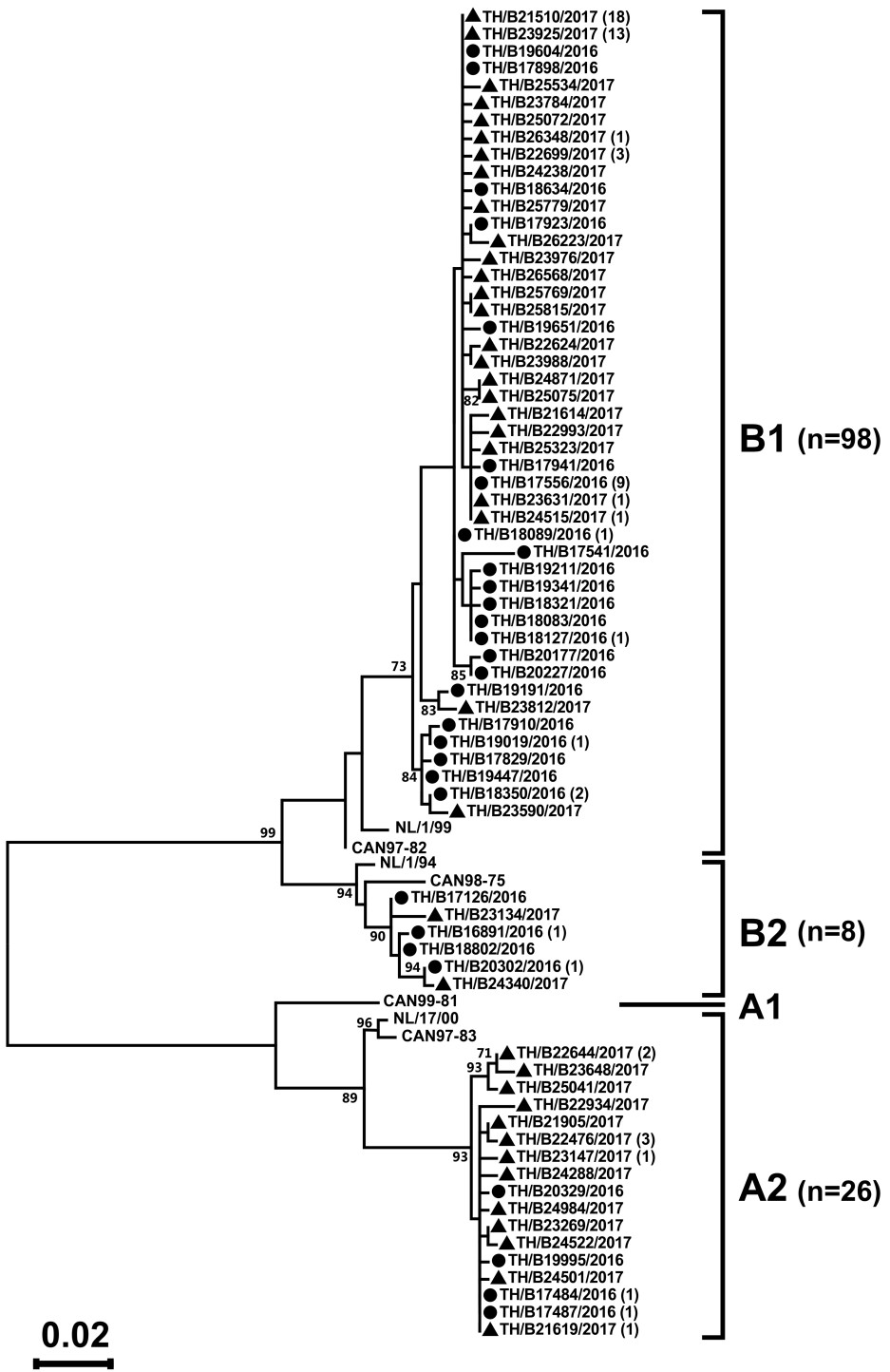

**Figure 3** **Phylogenetic analysis of hMPV subgroup A and B based on the partial nucleotide sequence of the F gene.** Tree was constructed using the maximum likelihood method and the Tamura–Nei model implemented in the MEGA6. The reliability of the tree was estimated using 1000 bootstrap pseudo-replicates. Bootstrap values > 70% are indicated at the branch nodes. Reference sequences for each genotype (A1, A2, B1, and B2) were obtained from GenBank. The scale bar represents the number of nucleotide substitutions per site between close relatives. Circles denote samples from Thailand 2016, while squares indicate strains from Thailand 2017. The number of strains are shown in parentheses.

**Table 4  Contribution of respiratory viruses as single or coinfections.**

| Virus | Total | Single infections | Double infections | Triple infections | Mixed infection (%) | RSV (non-typed) | RSV-A | RSV-B | hMPV (non-typed) | hMPV-A2 | hMPV-B1 | hMPV-B2 | Flu-H1N1 | Flu-H3N2 | Flu-B |
|---|---|---|---|---|---|---|---|---|---|---|---|---|---|---|---|
| RSV (non-typed) | 527 | 397 | 128 | 2 | **25.0** | – | 0 | 0 | 18 | 1 | 1 | 1 | 23 | 68 | 20 |
| RSV-A | 176 | 158 | 18 | 0 | **10.2** | 0 | – | 13 | 1 | 0 | 1 | 0 | 1 | 2 | 0 |
| RSV-B | 321 | 293 | 27 | 1 | **8.7** | 0 | 13 | – | 3 | 1 | 1 | 0 | 2 | 6 | 3 |
| hMPV (non-typed) | 186 | 161 | 22 | 3 | **13.4** | 18 | 1 | 3 | – | 0 | 0 | 0 | 1 | 3 | 2 |
| hMPV-A2 | 26 | 24 | 2 | 0 | **7.7** | 1 | 0 | 1 | 0 | – | 0 | 0 | 0 | 0 | 0 |
| hMPV-B1 | 98 | 93 | 5 | 0 | **5.1** | 1 | 1 | 1 | 0 | 0 | – | 0 | 1 | 0 | 1 |
| hMPV-B2 | 8 | 7 | 1 | 0 | **12.5** | 1 | 0 | 0 | 0 | 0 | 0 | – | 0 | 0 | 0 |
| Flu-H1N1 | 349 | 321 | 28 | 0 | **8.0** | 23 | 1 | 2 | 1 | 0 | 1 | 0 | – | 0 | 0 |
| Flu-H3N2 | 815 | 738 | 75 | 2 | **9.4** | 68 | 2 | 6 | 3 | 0 | 0 | 0 | 0 | – | 0 |
| Flu-B | 364 | 338 | 24 | 1 | **6.9** | 20 | 0 | 3 | 2 | 0 | 1 | 0 | 0 | 0 | – |

the alternating antigenic pattern of RSV infection over time (*Ohno et al., 2013*; *Hirsh et al., 2014*; *Fall et al., 2016*; *Auksornkitti et al., 2014*; *Thongpan et al., 2017*). It has been hypothesized that the periodic shifts in the predominant RSV subgroup are driven by the dynamics of population immunity and subgroup-specific herd immunity (*Botosso et al., 2009*). Regarding the relationship between clinical severity of infection and RSV types and subtypes, some studies have observed that RSV group A infection was associated with an increased illness severity (*McConnochie et al., 1990*; *Jafri et al., 2013*), while other studies observed that RSV group B infection resulted in more severe disease (*Hornsleth et al., 1998*; *Tran et al., 2013*). In the present study, the emerging genotypes of ON1 and BA9 completely replaced the previous genotypes, such NA1, and other BA genotypes as was found in other countries (*Dapat et al., 2010*; *Esposito et al., 2015*), although it has been observed that they do not appear to cause more severe disease than other genotypes (*Panayiotou et al., 2014*).

Phylogenetic analysis of the hMPV F gene in the present study showed that both A and B types co-circulated in Thailand over the two-year study period. Similar to our findings, all three subtypes of hMPV (A2, B1, and B2) co-circulated each year in other studies, including South Korea, Italy, Australia, and Norway (*Gerna et al., 2005*; *Mackay et al., 2006*; *Chung et al., 2008*; *Moe et al., 2017*). Although hMPV genotype A might be more virulent than genotype B (*Vicente et al., 2006*), data in the literature on the association between clinical symptoms and hMPV genotype remains unclear as some authors show a higher severity of illness, (*Vicente et al., 2006*; *Arnott et al., 2013*), while others did not (*Agapov et al., 2006*; *Manoha et al., 2007*). Furthermore, the prevalence of mostly influenza A H3N2 contrasts with the limited circulation of influenza B during this two-year study period. The predominance of influenza A H3N2 in 2016 was observed both in Thailand (*Suntronwong et al., 2017*) and the United States (*Blanton et al., 2017*).

Regarding multiple infections, RSV was co-detected mainly with influenza virus infection, which is consistent with an overlap of seasonal RSV and influenza virus infections. There have been reports showing no relationship between disease severity and multiple virus infections (*Lim et al., 2016*), while other studies have shown that viral co-infection was significantly associated with longer duration of symptoms, especially in RSV, and that this may increase the clinical severity of acute respiratory infection among children infected with RSV (*Cho et al., 2013*; *Harada et al., 2013*).

This study had several limitations. The convenient samples in this study may not be representative of the patient population in Bangkok. Since these samples were not tested for other respiratory viruses such as human parainfluenza virus and rhinovirus, we may have missed the identification of other respiratory pathogens. Samples were anonymized and had limited accompanying clinical data, therefore we were unable to examine the association between viral genotypes and clinical severity, although ILI clinical symptoms are generally similar regardless of viral etiology.

## CONCLUSION

Influenza viruses were the most common respiratory viruses diagnosed among ILI cases in this study. While RSV and hMPV infections were found mainly in young children and sporadically in adults, influenza virus infection was prevalent in adults and the elderly. A small number of individuals had dual infections, most commonly RSV and influenza A H3N2. Due to overlapping seasonal occurrence of these viral infections, accurate and rapid molecular detection can potentially assist clinicians and researchers in the treatment and surveillance to limit viral spread. The data presented here add to our understanding of the epidemiology of RSV, hMPV, and influenza causing respiratory illness in Thailand.

## ACKNOWLEDGEMENTS

We would like to thank the staff of Bangpakok 9 International Hospital in Bangkok for their technical and administrative assistance.

### Funding

This work was supported by the National Science and Technology Development Agency (NSTDA) for the Research Chair Grant (P-15-50004), Chulalongkorn University, the Center of Excellence in Clinical Virology (GLE 58-014-30-004, RES560530093) Chulalongkorn University and Chulalongkorn Hospital. This research was also supported by the 100th Anniversary Chulalongkorn University Fund for doctoral scholarship and the Overseas Research Experience Scholarship for Graduate Student to Ilada Thongpan. The funders had no role in study design, data collection and analysis, decision to publish, or preparation of the manuscript.

## Grant Disclosures

The following grant information was disclosed by the authors:

National Science and Technology Development Agency (NSTDA) for the Research Chair Grant: P-15-50004.

Chulalongkorn University, the Center of Excellence in Clinical Virology: GLE 58-014-30-004, RES560530093.

Chulalongkorn University and Chulalongkorn Hospital.

100th Anniversary Chulalongkorn University Fund for doctoral scholarship.

Overseas Research Experience Scholarship for Graduate Student to Ilada Thongpan.

## Competing Interests

The authors declare there are no competing interests.

## Author Contributions

- Ilada Thongpan conceived and designed the experiments, performed the experiments, analyzed the data, prepared figures and/or tables, authored or reviewed drafts of the paper, approved the final draft.
- Nungruthai Suntronwong and Preeyaporn Vichaiwattana performed the experiments, approved the final draft.
- Nasamon Wanlapakorn analyzed the data, prepared figures and/or tables, authored or reviewed drafts of the paper, approved the final draft.
- Sompong Vongpunsawad analyzed the data, prepared figures and/or tables, authored or reviewed drafts of the paper, approved the final draft.
- Yong Poovorawan conceived and designed the experiments, contributed reagents/materials/analysis tools, prepared figures and/or tables, authored or reviewed drafts of the paper, approved the final draft.

## Ethics

The following information was supplied relating to ethical approvals (i.e., approving body and any reference numbers):

The Institutional Review Board of the Faculty of Medicine of Chulalongkorn University approved this study (IRB number 609/59).

## Data Availability

The raw data files are available in Supplemental Files and in GenBank: MH447703–MH447725 (RSV-A), MH447726–MH447818 (RSV-B), and MH447819–MH447950 (hMPV).

## Supplemental Information

Supplemental information for this article can be found online at http://dx.doi.org/10.7717/peerj.6748#supplemental-information.

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

# PeerJ

**Jafri HS, Wu X, Makari D, Henrickson KJ. 2013.** Distribution of respiratory syncytial virus subtypes A and B among infants presenting to the emergency department with lower respiratory tract infection or apnea. *Pediatric Infectious Disease Journal* **32(4)**:335–340 DOI 10.1097/INF.0b013e318282603.

**Johnstone J, Majumdar SR, Fox JD, Marrie TJ. 2008.** Human metapneumovirus pneumonia in adults: results of a prospective study. *Clinical Infectious Diseases* **46(4)**:571–574 DOI 10.1086/526776.

**Lim FJ, De Klerk N, Blyth CC, Fathima P, Moore HC. 2016.** Systematic review and meta- analysis of respiratory viral coinfections in children. *Respirology* **21(4)**:648–655 DOI 10.1111/resp.12741.

**Mackay IM, Bialasiewicz S, Jacob KC, McQueen E, Arden KE, Nissen MD, Sloots TP. 2006.** Genetic diversity of human metapneumovirus over 4 consecutive years in Australia. *The Journal of Infectious Diseases* **193(12)**:1630–1633 DOI 10.1086/504260.

**Mahony JB. 2008.** Detection of respiratory viruses by molecular methods. *Clinical Microbiology Reviews* **21(4)**:716–747 DOI 10.1128/CMR.00037-07.

**Manoha C, Espinosa S, Aho SL, Huet F. 2007.** Epidemiological and clinical features of hMPV, RSV and RVs infections in young children. *Journal of Clinical Virology* **38(3)**:221–226 DOI 10.1016/j.jcv.2006.12.005.

**McConnochie KM, Hall CB, Walsh EE, Roghmann KJ. 1990.** Variation in severity of respiratory syncytial virus infections with subtype. *Jornal de Pediatria* **117**:52–62.

**Moe N, Krokstad S, Stenseng IH, Christensen A, Skanke LH, Risnes KR, Nordbø SA, Døllner H. 2017.** Comparing human metapneumovirus and respiratory syncytial virus: viral co-detections, genotypes and risk factors for severe disease. *PLOS ONE* **12(1)**:e0170200 DOI 10.1371/journal.pone.0170200.

**Murray EL, Klein M, Brondi L, McGowan JE, Van Mels Jr C, Brooks WA, Kleinbaum D, Goswami D, Ryan PB, Bridges CB. 2012.** Rainfall, household crowding, and acute respiratory infections in the tropics. *Epidemiology and Infection* **140(1)**:78–86 DOI 10.1017/s0950268811000252.

**Ohno A, Suzuki A, Lupisan S, Galang H, Sombrero L, Aniceto R, Okamoto M, Saito M, Fuji N, Otomaru H, Roy CN, Yamamoto D, Tamaki R, Olveda R, Oshitani H. 2013.** Genetic characterization of human respiratory syncytial virus detected in hospitalized children in the Philippines from 2008 to 2012. *Journal of Clinical Virology* **57(1)**:59–65 DOI 10.1016/j.jcv.2013.01.001.

**Panayiotou C, Richter J, Koliou M, Kalogirou N, Georgiou E, Christodoulou C. 2014.** Epidemiology of respiratory syncytial virus in children in Cyprus during three consecutive winter seasons (2010–2013): age distribution, seasonality and association between prevalent genotypes and disease severity. *Epidemiology and Infection* **142(11)**:2406–2411 DOI 10.1017/s0950268814000028.

**Parsania M, Poopak B, Pouriayevali MH, Haghighi S, Amirkhani A, Nateghian A. 2016.** Detection of human metapneumovirus and respiratory syncytial virus by real-time polymerase chain reaction among hospitalized young children in Iran. *Jundishapur Journal of Microbiology* **9(3)**:e32974 DOI 10.5812/jjm.32974.

**Paynter S, Ware RS, Sly PD, Weinstein P, Williams G. 2015.** Respiratory syncytial virus seasonality in tropical Australia. *Australian and New Zealand Journal of Public Health* **39(1)**:8–10 DOI 10.1111/1753-6405.12347.

**Richter J, Panayiotou C, Tryfonos C, Koptides D, Koliou M, Kalogirou N, Georgiou E, Christodoulou C. 2016.** Aetiology of acute respiratory tract infections in hospitalised children in cyprus. *PLOS ONE* **11(1)**:e0147041 DOI 10.1371/journal.pone.0147041.

**Suntronwong N, Klinfueng S, Vichiwattana P, Korkong S, Thongmee T, Vongpunsawad S, Poovorawan Y. 2017.** Genetic and antigenic divergence in the influenza A(H3N2) virus circulating between 2016 and 2017 in Thailand. *PLOS ONE* **12(12)**:e0189511 DOI 10.1371/journal.pone.0189511.

**Suwannakarn K, Payungporn S, Chieochansin T, Samransamruajkit R, Amonsin A, Songserm T, Chaisingh A, Chamnanpood P, Chutinimitkul S, Theamboonlers A, Poovorawan Y. 2008.** Typing (A/B) and subtyping (H1/H3/H5) of influenza A viruses by multiplex real-time RT-PCR assays. *Journal of Virological Methods* **152(1–2)**:25–31 DOI 10.1016/j.jviromet.2008.06.002.

**Tamura K, Stecher G, Peterson D, Filipski A, Kumar S. 2013.** MEGA6: molecular evolutionary genetics analysis version 6.0. *Molecular Biology and Evolution* **30(12)**:2725–2729 DOI 10.1093/molbev/mst197.

**Thanasugarn W, Samransamruajkit R, Vanapongtipagorn P, Prapphal N, Van den Hoogen B, Osterhaus AD, Poovorawan Y. 2003.** Human metapneumovirus infection in Thai children. *Scandinavian Journal of Infectious Diseases* **35(10)**:754–756 DOI 10.1080/00365540310000094.

**Thongpan I, Mauleekoonphairoj J, Vichiwattana P, Korkong S. 2017.** Respiratory syncytial virus genotypes NA1, ON1, and BA9 are prevalent in Thailand, 2012–2015. *PeerJ* **5**:e3970 DOI 10.7717/peerj.3970.

**Tran DN, Pham TM, Ha MT, Tran TT, Dang TK, Yoshida LM, Okitsu S, Hayakawa S, Mizuguchi M, Ushijima H. 2013.** Molecular epidemiology and disease severity of human respiratory syncytial virus in Vietnam. *PLOS ONE* **8(1)**:e45436 DOI 10.1371/journal.pone.0045436.

**Venter M, Madhi SA, Tiemessen CT, Schoub BD. 2001.** Genetic diversity and molecular epidemiology of respiratory syncytial virus over four consecutive seasons in South Africa: identification of new subgroup A and B genotypes. *Journal of General Virology* **82(9)**:2117–2124 DOI 10.1099/0022-1317-82-9-2117.

**Vicente D, Montes M, Cilla G, Perez-Yarza EG, Perez-Trallero E. 2006.** Differences in clinical severity between genotype A and genotype B human metapneumovirus infection in children. *Clinical Infectious Diseases* **42(12)**:e111-3 DOI 10.1086/504378.

**Weber MW, Mulholland EK, Greenwood BM. 1998.** Respiratory syncytial virus infection in tropical and developing countries. *Tropical Medicine & International Health* **3(4)**:268–280 DOI 10.1046/j.1365-3156.1998.00213.x.

**Williams JV, Harris PA, Tollefson SJ, Halburnt-Rush LL, Pingsterhaus JM, Edwards KM, Wright PF, Crowe Jr JE. 2004.** Human metapneumovirus and lower respiratory tract disease in otherwise healthy infants and children. *New England Journal of Medicine* **350(5)**:443–450 DOI 10.1056/NEJMoa025472.

**World Health Organization (WHO). 2018.** Influenza (Seasonal). *Available at http: //www.who.int/mediacentre/factsheets/fs211/en* (accessed on 16 April 2018).

**Zhang D, He Z, Xu L, Zhu X, Wu J, Wen W, Zheng Y, Deng Y, Chen J, Hu Y, Li M, Cao K. 2014.** Epidemiology characteristics of respiratory viruses found in children and adults with respiratory tract infections in southern China. *International Journal of Infectious Diseases* **25**:159–164 DOI 10.1016/j.ijid.2014.02.019.