# Peer review of "Respiratory syncytial virus, human metapneumovirus, and influenza virus infection in Bangkok, 2016-2017"

_PeerJ, doi:10.7717/peerj.6748_

## Round 0.1 · original submission · Minor Revisions

Two expert reviewers have commented on your paper, and I look forward to reading your revised submission and response document.

Please note the journal's requirement that the data be made available. One of the referees pointed this out and this is a requirement that I will be confirming in the next round.

Reviewer 1 ·

Basic reporting

There are a few places where the English needs correcting.

Some of the references could be more relevant.

For examples, see specific comments below in General comments.

Experimental design

Would be important to include a few additional details about the methods of the epidemiologic study design. For specifics see below in General comments.

Validity of the findings

Generally fine given the limitations of the study design. Would be critical to include a limitations paragraph in the discussion. For specifics see below in General comments.

Additional comments

This paper describes the prevalence and genetic characteristics of several respiratory viruses in a convenience sample of outpatients seeking care at one hospital in Bangkok. The findings are interesting and contribute to the general knowledge of influenza viruses, RSV and human metapneumovirus from a tropical country. However, they may not be representative of the broader population in Bangkok.

(Pages were not numbered so I used the pdf page number as reference.)
Page 7, line 53. Transcriptase is the enzyme. The test is called “reverse transcription…”
Page 7, lines 54-56. You are implying that the PCR was/should be used for clinical care. However, in this study you used it for surveillance/research and did it retrospectively on batched samples (I think). It is ok to say it may have utility in clinical care (with the caveat that it might be expensive), but I think you should also say it is useful for surveillance purposes. Also, you do not come back to this point about clinical care in the discussion. I would delete.
Page 8, specimen section. Please include more information on the methods. Were these from both outpatient and inpatient populations (I assume just outpatients)? What is the catchment population of this hospital? Did you have other clinical or epidemiologic data? Did you enroll persons of all ages? Was there any sampling methodology? Were all persons sampled or only, say, the first X of the day. You say it was a convenience sample, but a little more information would help the reader to understand the biases inherent in your sample.
Page 11, Beginning of results. Please add a sentence of two about your sample before presenting the viral results. How many people, what was the age/sex distribution, etc. Can you say anything about refusals?
Page 11, line 143. Please do not refer to this as incidence as it is just prevalence. Was there any seasonal difference in RSV A vs. B?
Page 13, line 173. It doesn’t look like you did any statistical testing, so you can’t say if any co-infections occurred more than chance. I would delete that sentence and just state what you found. It would be nice to have percentages in Table 4.
Page 14, line 183. You say “cost-effectively” but you present no data to support that claim. I would delete it.
Page 14, line 187. You need to insert the word “prevalence” after RSV and hMPV. The it would be “was” not “were.” Also, delete the word “at” before the word between.
Page 14, line 190. It seems odd that you only reference one paper from Iran. I would be more comprehensive or maybe just cite other papers from Thailand.
Page 15, line 213. You can’t say it is consistent with other countries and not consistent with other countries. These two statements are inconsistent. In fact, if your argument is that RSV type varies by year you should lead with that. Then, if you compare to other countries you probably should compare the same seasons.
Page 16, line 225. You cannot say anything about severity from your study since I think you only had outpatients. If you are referring to what other have found, please be clear. Otherwise you should delete that part.
Page 16, line 232. You need to say “However, a few studies…”
Page 16, line 237. Again, please use the word prevalence and not incidence. Also, the dominant influenza A subtype varies by year, so you need to be when you say H3 dominated what year that was. Also, there are better references to summarize the influenza activity in the US.
Page 16. You need to add a limitations paragraph. Please at least include 1) convenience sample so probably not representative of all outpatient respiratory illness in Bangkok, 2) only outpatients so not able to investigate the questions raised about severity, and 3) ILI clinical syndrome may not be the best to identify RSV.
Page 17, 255. I think the main point should be that influenza viruses were the most frequently detected viruses in your sample.
Figure 1C. For the influenza graph, I would add a link for percent positive for any flu virus. It would be nice if you could say how the data are helpful to clinicians and researchers.
Table 3. I the p-value for the age distribution? It is not clear in the table. I don’t really think the p-value is that helpful. It only tells you that there are differences overall, but not which difference are important. Also, did you have any patients 65 years or older? That is the standard cut-off for influenza so would be important to know. For sex, you don’t need to list both male and female.
I was not able to find the raw data on the website. Please be sure that is available.

Reviewer 2 ·

Basic reporting

.

Experimental design

.

Validity of the findings

.

Additional comments

Thongpan et al. have examined over 8,000 respiratory specimens from patients with respiratory illnesses for the presence of respiratory syncytial virus (RSV), human metapneumovirus (hMPV) and influenza virus throughout two recent years in Bangkok, Thailand. Influenza-like illness peak each year during the rainy season. The investigators were able to detect one or more of these viruses by multiplex real-time reverse transcription PCR in 30% of NP swab samples. PCR products for the two major subtypes of RSV and hMPV, and the two currently circulating types of influenza virus were sequenced. Influenza virus was most prevalent overall, and in the older age groups, while RSV and hMPV were more prevalent in young children, though also present in all age groups.

This report is a thorough investigation of three prominent respiratory viruses in a large population in a densely populated region over two entire years. The number of influenza virus and RSV viruses identified were >1,000, and hMPV >300, an excellent sample size for defining the relative numbers of these infections and when they occurred.

The authors used samples from patients with influenza-like illnesses (l.63). Does this mean that the RSV and hMPV cause influenza-like symptoms?

The authors do not discuss why they were only able to detect a virus in ~30% of the samples. How were these samples handled, stored? What was the efficiency of detection for other similar studies? Was their method less sensitive than expected? Are there other respiratory viruses that could have caused these ILI symptoms that were not tested for? Other influenza viruses, parainfluenza viruses, rhinoviruses, adenoviruses or others? Were any of these ruled out?

The frequency of co-infections seems low, 3% or less in all age groups. Could that be because they did not test for many other respiratory pathogens? The authors should state the frequency of co-infection as co-infection with two or more of the 6 virus types tested for.

It seems surprising all 23 RSV-A strains were the same genotype, ON1, and all 93 of the RSV-B strains were BA9. Would the primers used be able to detect all the other RSV-A and RSV-B strains?

Figure 1 shows an excellent plot of the overall results of the monthly detection of RSV, hMPV and influenza virus plotted on the backdrop of the seasonal distribution of each virus over the two years of study. It is obvious from the plot that in the first year, the RSV peak was on the leading edge of the overall respiratory virus plot, and hMPV was on the trailing edge of the peak. The hMPV also remained high for three months beyond the overall peak. Influenza virus coincided very well with the overall peak (because there were more influenza that RSV and hMPV combined). In contrast, in the second year, all three viruses peaked simultaneously during the overall peak. Surprisingly, the authors did not mention any of this. They need to discuss this interesting result and what it means for predicting the virus that a patient is infected by the most recent detections, which they do discuss.

The sequencing method and the primers used for sequencing are not mentioned, but should be.

In Figures 2 and 3, which viruses were determined in this study? Are they the ones with the circle or triangle symbols? Why triangle, circle?

Suggestions:
l.39. contributor to morbidity
l.188. 15.4 and 21.2%...5.5 and 5.7%
l.191. lower burden
l.194. aged 13-18 years
l.202. settings such as Indonesia…New Zealand. In these regions, respiratory infections peak in the rainy season and decline during the hot and dry months.
l.232. However, other studies have reported no clinical differences between hMPV genotypes
L.234/5. What does this sentence mean? It implies that we now know the relationship between clinical severity and genetic variability of hMPV, but does tell the reader what that is, and ends with “has remained unclear”.
l.240. flu subtype(S, 2017), similar to reports from the US
l.247. virus consistent with an overlap of seasonal RSV and influenza virus infections.
l.255. The conclusion section could be improved. See attached pdf.
Table 2 title. hMPV or influenza virus.
In Table 4, the # for Dual infections does not the sum of the dual infections in the other two virus columns (Influenza dual is 127; Inf+RSV is 125; Inf+hMPV is 8: 127 does not equal 125 + 8) Adding in the triple infections does not help. Otherwise, a very useful table.

---

## Round 0.2 · Minor Revisions

Before I send this back to the referees for their re-review, I would like wrap-up the business of data availability.

All the genetic sequences seem to be available to me, but where are the data files for Figure 1? Please forgive me if I missed them, but I cannot seem to find them in your SI.

Thank you. Once you make this available (or clarify to me where the data are, if you have already provided it), I will send the manuscript for re-review.

There is no need to modify the manuscript.

---

## Round 0.3 · accepted · Accept

One of the original reviewers has reviewed your resubmitted manuscript, while the other one declined. Based on the comments of the reviewer as well as my own read, I am pleased to accept your paper.

Please note that the referee made some small suggestions; please implement these.

# Reviewer 2 ·

Basic reporting

No comment

Experimental design

No comment

Validity of the findings

No comment

Additional comments

l.23. a word seems to be missing: ...metropolitan ______ of Thailand... Regoin? District?
l.188. a previous study...
l.230. while other authors found a higher...
l.245. parainfluenza viruses and rhinoviruses, we... Could adenovirus also cause ILI?

I agree that finding different peaks for the 3 viruses the first year and not the second does not establish a pattern. But it does suggest separate introductions of each virus to the community.